# No evidence for ongoing replication on ART in SIV-infected macaques

Taina T. Immonen[1], Christine M. Fennessey[1], Leslie Lipkey[1], Laura Newman[1], Agatha Macairan[1], Marjorie Bosche[1], Nora Waltz[1], Gregory Q. Del Prete[1], Jeffrey D. Lifson[1] & Brandon F. Keele [1] ✉

The capacity of HIV-1 to replicate during optimal antiretroviral therapy (ART) is challenging to assess directly. To gain greater sensitivity to detect evolution on ART, we used a nonhuman primate (NHP) model providing precise control over the level of pre-ART evolution and more comprehensive analyses than are possible with clinical samples. We infected 21 rhesus macaques (RMs) with the barcoded virus SIVmac239M and initiated ART early to minimize baseline genetic diversity. RMs were treated for 285–1200 days. We used several tests of molecular evolution to compare 1352 near-full-length (nFL) SIV DNA single genome sequences from PBMCs, lymph nodes, and spleen obtained near the time of ART initiation and those present after long-term ART, none of which showed significant changes to the SIV DNA population during ART in any animal. To investigate the possibility of ongoing replication in unsampled putative tissue sanctuaries during ART, we discontinued treatment in four animals and confirmed that none of the 336 nFL SIV RNA sequences obtained from rebound plasma viremia showed evidence of evolution. The rigorous nature of our analyses reinforced the emerging consensus of a lack of appreciable ongoing replication on effective ART and validates the relevance of this NHP model for cure studies.

Current antiretroviral therapy (ART) successfully suppresses HIV replication by blocking the infection of new cells, leading to a rapid decline in HIV plasma RNA concentrations to levels that are undetectable by clinical assays[1–3]. However, ART does not target integrated HIV-1 DNA, allowing the virus to persist during therapy in a rebound-competent viral reservoir (RCVR) primarily comprising resting, long-lived memory CD4+ T cells, including expanded CD4+ T cell clones harboring clonally integrated proviruses[4–9]. If treatment is discontinued, reactivation of quiescent infected cells nearly invariably rekindles viral replication and leads to systemic spread, requiring lifelong adherence to ART[10,11]. Understanding how HIV-1 infectivity is sustained during ART is critical for the design of treatment strategies that would allow individuals living with HIV-1 to safely discontinue therapy without viral recrudescence.

There are at least two distinct but not mutually exclusive possible mechanisms that could contribute to the size of the established RCVR while on continuous long-term ART: proliferation/regression of CD4+ T cells that were infected prior to ART initiation, and/or ongoing rounds of active viral replication below the limit of detection in plasma, potentially in tissue sanctuaries with poor drug penetration[12]. Recent studies have demonstrated that clonally expanded HIV-infected cells comprise a large fraction of the persisting HIV population and can be a source of viral rebound, highlighting the importance of clonal expansion for maintaining the RCVR during ART[9,13–22]. In contrast, the extent to which any ongoing rounds of replication contribute to viral persistence is challenging to quantify directly[12,23–30]. Importantly, ongoing replication, even at a level not detectable in plasma, could have clinical consequences and lead to the accumula-

[1]AIDS and Cancer Virus Program, Frederick National Laboratory for Cancer Research, Frederick, MD 21702, USA. ✉e-mail: keelebf@mail.nih.gov

tion of new mutations in the persisting virus population, which could hinder the efficacy of interventions aimed at harnessing host immune responses to achieve virological control after ART discontinuation. Optimizing treatment strategies therefore requires determining if, and how much, ongoing viral replication contributes to viral persistence during long-term ART.

While it is not possible to directly quantify active HIV-1 replication in tissues during suppressive ART, the accumulation of genetic changes in longitudinal viral sequences can serve as a marker of ongoing replication. Detecting a signal of viral evolution requires being able to sample newly emergent viral variants from the vast persisting HIV-1 population and correctly distinguish genetic changes acquired during ART from those present in the virus population prior to ART initiation. Several studies have analyzed the evolution of viral RNA or DNA sequences obtained from longitudinal peripheral blood samples, none of which demonstrated genetic changes consistent with ongoing cycles of replication during ART, including those from individuals starting ART early in infection with limited pre-ART diversity[16,25,27,30,31]. However, a stable HIV-1 population in blood during long-term ART does not preclude ongoing viral replication in putative, compartmentalized sanctuary sites in tissues, as newly emergent variants may not be readily detectable in the periphery.

Studies assessing HIV-1 replication in tissues have yielded contradictory results, with some reporting evidence of ongoing replication in cerebrospinal fluid and lymphoid tissues, including the spleen and gut-associated lymphoid tissue[12,26,28,29]. However, the apparently continued replication in lymph nodes (LNs) found in Lorenzo-Redondo et al[26]. has been contested based on methodological grounds, including the short duration of ART and the phylogenetic approach used to detect evolution[32,33]. In contrast, more recent studies using more accurate sequencing techniques found no genetic changes to viral sequences obtained from rectal and LN biopsies and diverse tissues collected at autopsy[18,22–24]. The consensus view emerging from these studies and informing current cure strategies assumes that optimal ART is fully suppressive. However, it is impossible to directly rule out replication in an unknown sanctuary site, because comprehensively assessing all potential sites of replication in tissues is not feasible. If low-level replication persists during suppressive ART, treatment discontinuation may allow actively replicating lineages to spread systemically and contribute to virus recrudescence, providing an alternative means to interrogate continued replication in putative sanctuary sites that are not possible to directly sample. One study characterizing the origins of viral recrudescence during ATI found that rebound virus was not more evolved than pre-therapy virus, however, the sensitivity of the phylogenetic approach used to detect evolution was likely limited by study participants initiating therapy in chronic HIV-1 infection[34].

Nonhuman primate (NHP) models of HIV-1 infection provide several advantages for studying ongoing replication on ART, including enabling frequent sampling of blood with precisely defined timing relative to initial infection, access to tissues via biopsy or at necropsy, early ART initiation to minimize baseline genetic diversity, an identical T/F virus with known genetic constituents in all animals, and the ability to use custom viruses not possible in humans. Although previous studies assessing ongoing replication during ART in simian immunodeficiency virus (SIV) or simian/human immunodeficiency virus (SHIV) infected NHPs did not yield evidence of continued viral evolution[35–40], therapy was initiated during chronic infection and/or the initial inoculum virus was genetically heterogeneous in these studies, making it more challenging to discriminate any new genetic changes from pre-existing diversity.

To overcome many of the challenges in detecting continued low-level HIV-1 replication during suppressive ART, we applied an NHP model of HIV-1 infection utilizing a genetically barcoded viral inoculum and near full-length (nFL) single genome sequencing (SGS) to track

changes to the persisting SIV population during ART at the resolution of individual mutations in 21 RMs. To simultaneously achieve a genetically homogeneous pre-ART viral population and the ability to track evolution within individual lineages, the animals were infected with the barcoded virus SIVmac239M[41–45], and ART was initiated early in infection to prevent the accumulation of genetic changes. To look for molecular evolution at the population level, we assessed if viral sequences obtained after long-term ART differed significantly from those obtained shortly after ART initiation. Comparative analysis of 1352 nFL sequences obtained from longitudinal on-ART PBMC samples, and in some animals, lymphatic tissue biopsies, did not yield any evidence that additional genetic changes occurred in the persisting virus population during long-term ART. To further rule out the possibility of ongoing low-level replication in an unsampled tissue sanctuary, we discontinued treatment in four animals and assessed whether any rebounding lineages showed increased evolution. During off-ART viral rebound, there was no evidence that plasma viral RNA had acquired additional mutations compared to the pre-ART virus population, indicating that viral rebound did not originate from viral lineages that had continued to replicate during ART. The rigorous nature of our study, which allowed unprecedented analytical depth and resolution into the virus population and more comprehensive analyses than are feasible with clinical samples, reinforces both the emerging consensus of no HIV-1 replication on effective ART[16,22–25,30,31] and validates the relevance of this NHP model for cure intervention studies[41,43,44].

## Results

### No evidence for SIV evolution during ART in PBMC in macaques treated at 27 dpi

Using various analytical approaches, we assessed if the SIV DNA population continued to evolve during ART in animals treated at 27 dpi (Fig. 1, cohort 1) compared to untreated control animals. Specifically, we used several tests for molecular evolution to determine if the sequences obtained after ART showed significant changes compared to sequences obtained shortly after ART initiation in (i) genetic divergence from the T/F sequence, (ii) genetic diversity, measured as average pairwise distance (APD), (iii) population genetic structure measured using a test for panmixia[23,25,30], and (iv) a positive evolutionary rate between samples, measured as the number of nucleotide substitutions per site over time. From cohort 1, we obtained a total of 111 nFL SIV DNA sequences from PBMCs at the first time point and 108 nFL SIV DNA sequences 289–251 days later. We found a fraction (11%, $n = 25$) of genomes that had obvious defects (large deletions >300 bp, frameshift or stop codon mutations, and hypermutated sequences), which were excluded from analyses. Additionally, we excluded individual sites where at least one sequence had a potential APOBEC-mediated mutation. Initiating ART relatively early in acute infection limited both baseline genetic diversity (median APD of 0.030%, range of 0.026–0.036%) and divergence from the founder virus (median $p$-distance of 0.016%, range of 0.013–0.018%). Rather than accumulating more genetic changes over time, SIV DNA populations at later time points were genetically closer to the founder sequence than at the time of ART initiation in each animal (median APD of 0.024%, range of 0.022–0.027%; median $p$-distance of 0.012%, range of 0.011–0.014%) although the observed reductions in genetic divergence and diversity were not statistically significant in individual animals. Consistent with this trend towards reduced genetic diversity and divergence, all animals had negative evolutionary rates during ART and the regression slopes were not significantly different from zero (Fig. 2A; Table 1). Finally, a test for panmixia revealed that the SIV DNA populations did not shift significantly during ART in four animals. Interestingly, the proportion of sequences with no mutations increased almost 3-fold during ART in one animal (2/14 to 8/20), resulting in non panmictic viral populations between the two-time points ($p = 0.04$), However,

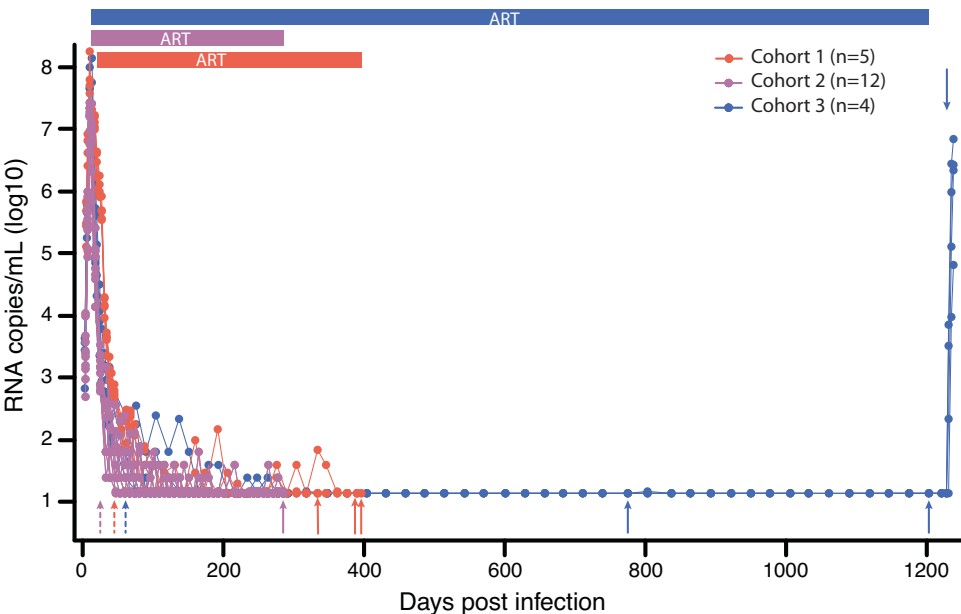

**Fig. 1 | Plasma viral load curves and sampling dates from 3 study cohorts.** Colored horizontal bars indicate the timeframe of ART treatment in each cohort. Dashed and solid arrows indicate when baseline and long-term ART SIV DNA samples were collected for each cohort, respectively. PVL has a 15 copy/mL limit of detection. Source data are provided as a Source Data file.

this difference is not explained by the emergence of new viral variants, but rather by a loss of variants and reversion towards the T/F sequence. We repeated the analysis with APOBEC-specific context mutations included which did not significantly impact the results (Supplement Table 1).

In two untreated control animals, a total of 22 and 32 nFL SIV RNA sequences were obtained from longitudinal plasma samples taken 14–17 days apart in each animal just after peak pVL (Supplemental Fig. 1A). Both animals showed detectable viral evolution, with significant increases in genetic diversity and/or genetic divergence from the founder sequence (APD of 0.046–0.075%; $p = 0.074$ and 0.024–0.061%; $p = 0.0057$, bootstrap test, and $p$-distance of 0.029–0.054%; $p = 6.1E-4$ and 0.019–0.049%; $p = 1.2E-03$; Wilcoxon paired sum test). The animals also had nearly identical positive evolutionary rates of 6.6E-3 and 6.5E-3 mutations per site per year (Table 1), in line with previous estimates[37,46]. In addition, the test for panmixia indicated that the plasma virus populations differed significantly between time points in each of the untreated animals ($p = 9.6E-3$ and $p = 2.0E-3$, respectively). Repeating the analysis with APBOEC-specific context mutations included did not significantly impact the results (Supplement Table 1).

Phylogenetic trees were constructed for each RM in cohort 1 and the displayed branching patterns were consistent with early immune selection, with at least one mutation shared by multiple sequences in each animal (Fig. 2B). Importantly, all major clades contained sequences from both time points, and the longest branches were associated with sequences obtained shortly after ART began, indicating that no additional viral mutations were detectable after ART was initiated. However, although the baseline genetic diversity was low in these animals compared to clinical cohorts, we observed polymorphisms that appeared to be under selection at the first sample time point, followed by a shift towards a more ancestral population during ART (Supplemental Fig. 2). In contrast, phylogenetic trees from the two untreated control animals showed clear evidence of evolution, with sequences from later time points having longer branches and clustering together based on shared mutations (Supplemental Fig. 1B, C). Overall, viral replication was readily detectable in the

control animals by each test for molecular evolution but undetectable in the ART-treated animals.

## No evidence for SIV evolution during ART in PBMC in macaques treated at 10 dpi

To evaluate viral populations with an even greater degree of baseline homogeneity prior to ART, we repeated our analysis for RMs starting ART at 10 dpi (Fig. 1, cohort 2). From cohort 2, we obtained a total of 263 nFL SIV DNA sequences from PBMCs at the first time point and 184 nFL SIV DNA sequences 260 days later. We found that a fraction (14%, $n = 64$) of genomes had obvious defects (large deletions >300 bp, frameshift or stop codon mutations, and APOBEC-induced hypermutation), which were excluded from analyses. Additionally, we excluded individual sites where at least one sequence had a potential APOBEC-mediated mutation. Initiation of ART prior to peak PVL resulted in a highly homogenous starting viral population with low genetic diversity and divergence (median APD of 0.012% and median $p$-distance of 0.0058%), which did not increase during 260 days on ART (median APD of 0.012% and $p$-distance of 0.0055%; Supplement Table 1). Consistent with these results, the evolutionary rate during ART was not significantly different from zero in any animal, and the SIV DNA populations were panmictic between time points (Supplemental Fig. 3). We repeated the analysis with APOBEC-specific context mutations included which did not significantly impact the results (Supplement Table 1).

To identify evolutionary relationships between individual sequences, we constructed a NJ-phylogeny combining all sequences from the 12 RMs and rooted on the SIVmac239 T/F sequence (Fig. 3). Overall, the sequences were well described by a star phylogeny, consistent with a Poisson mutation process in rapidly expanding populations without selection. The longest branches encompassed sequences from both time points, with only seven sequences in total harboring more than two mutations. The longest branch in the tree had six mutations, five of which were embedded within APOBEC motifs and likely acquired during a single replication cycle. The tree had little internal structure, with only nine clusters of two to four sequences, each sharing a single mutation. Interestingly, five of these clusters

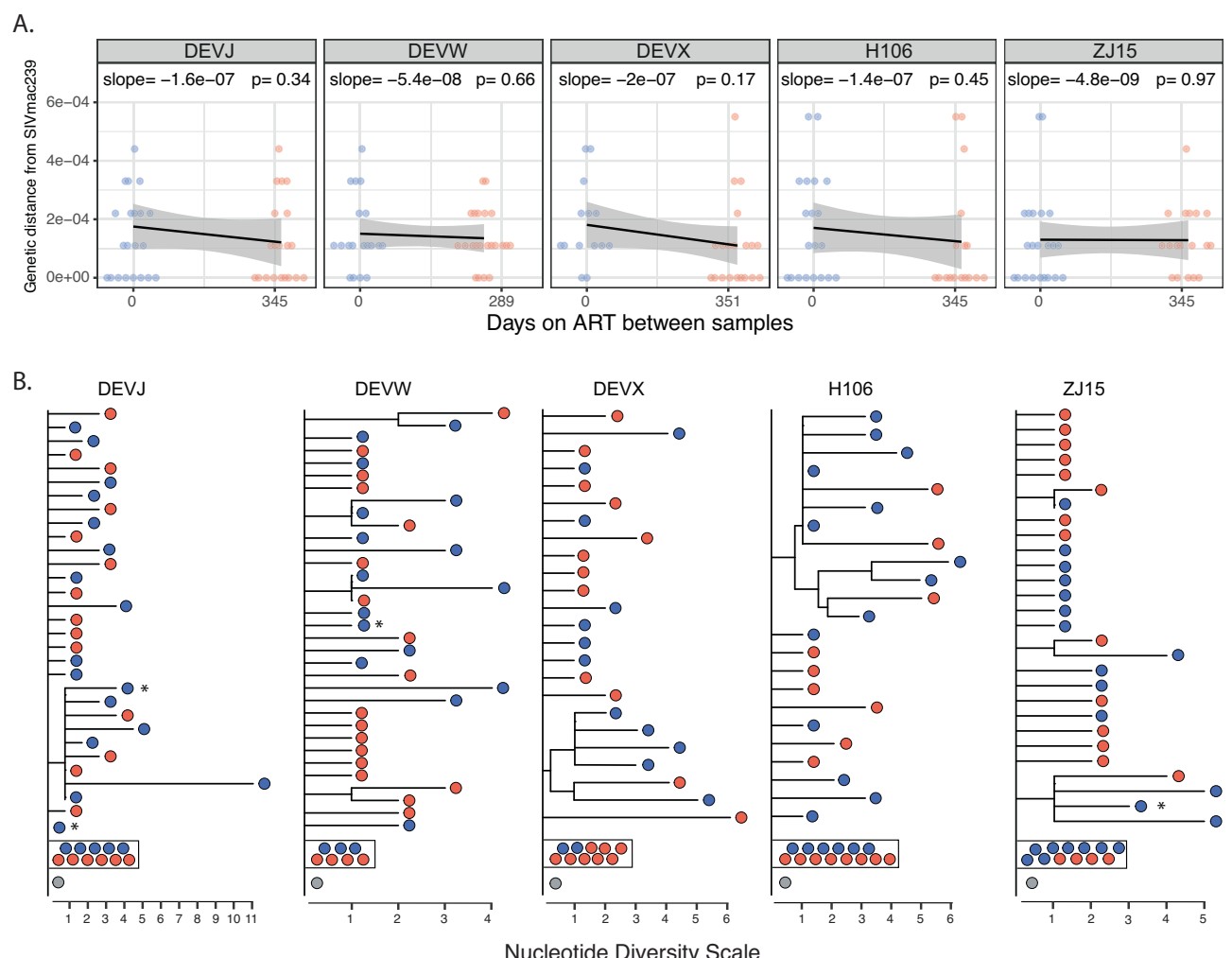

**Fig. 2 | No evidence for evolution in cohort 1 animals initiating ART at 27 dpi.** The scatter plot at each time point shows the distribution of *p*-distances from the SIVmac239 founder with black lines corresponding to the linear regression slopes (mutations per site per day) and the grey bands to the 95% confidence intervals around the slope (**A**). The evolutionary rates during ART are not significantly different from zero in any animal (two-sided *t*-test). The phylogenetic trees show intermixing of nFL SIV DNA sequences obtained from PBMCs at 45 dpi (blue) and 285–345 dpi (red) in each animal (**B**). Asterisks indicate sequences with short indels. Source data are provided as a Source Data file.

contained sequences from two different animals and all were non-synonymous mutations.

We next assessed if the characteristics of individual mutations differed between sequences obtained from different time points. There were 242 nucleotide substitutions, including ones with APO-BEC signatures, among the 383 nFL sequences generated for the dataset. Overall, transition mutations were 8-fold more frequent than transversions, with no significant differences in their distribution between time points (*p* = 0.2; Fisher's exact test). G-to-A substitutions were the most common nucleotide change at each time point. However, most of these mutations occurred at sites that were embedded in nucleotide contexts indicative of APOBEC mediation (56% at 25 dpi and 58% at 285 dpi), despite the exclusion of heavily hypermutated sequences from the analysis. Nonsynonymous mutations were strongly underrepresented, constituting only 118 out of 208 mutations in non-overlapping coding regions, and the proportion of nonsynonymous mutations did not differ significantly between time points. Overall, among the over three million bases examined, we did not find any genetic changes in the persisting SIV DNA PBMC population consistent with ongoing cycles of replication during ART.

## No evidence for SIV evolution during ART in LNs or PBMCs in macaques treated at 10 dpi

To extend our analyses to include viral populations in key tissue sites of SIV replication in the setting of longer durations of ART, we examined viral DNA sequences in tissues from four RMs infected with SIV-mac239M and treated with ART for 3 years starting at 10 dpi (Fig. 1, cohort 3). From these RMs, we obtained a total of 310 nFL SIV DNA sequences from PBMCs at baseline, 439 nFL SIV DNA sequences from tissues 714 days later, and 103 nFL SIV DNA sequences from PBMCs 1142 days from baseline. We found that a fraction (9%, *n* = 77) of genomes had obvious defects (large deletions >300 bp, frameshift and stop codon mutations and hypermutated sequences), which were excluded from analyses. Additionally, we excluded individual sites where at least one sequence had a potential APOBEC-mediated mutation. Baseline measures of the SIV DNA PBMC population (median APD of 0.012% and *p*-distance of 0.00059%) were nearly identical to those observed in cohort 2 highlighting the consistency of viral replication provided by this virus model. We compared these baseline viral sequences to ones obtained from tissues after 714 days on ART and from PBMCs after 1142 days on ART. The genetic diversity and divergence of the on-ART viral populations did not change significantly

**Table 1 | Evolutionary analyses excluding potential APOBEC mutations**

| Group | Animal | Day | Tissue | Divergence | Diversity | Panmixia | Mutations site⁻¹ day⁻¹ (95% CI) | Pr(>\|t\|) | df |
|---|---|---|---|---|---|---|---|---|---|
| 0 | 7811 | 17 | PBMC | **1.23E-03**[a] | **5.70E-03** | **2.00E-03** | **1.79E-05 (1.00E-05–2.57E-05)** | **1.27E-04** | 20 |
| 0 | 2561I | 14 | PBMC | **6.15E-04** | 0.08 | **0.01** | **1.81E-05 (7.49E-06–1.69E-05)** | **1.53E-03** | 30 |
| 1 | DEVJ | 345 | PBMC | 0.43 | 0.37 | 0.62 | -1.57E-07 (-4.83E-07–2.87E-05) | 0.34 | 40 |
| 1 | DEVW | 289 | PBMC | 0.79 | 0.70 | 0.86 | -5.38E-08 (-2.99E-07–1.92E-07) | 0.66 | 39 |
| 1 | DEVX | 351 | PBMC | 0.07 | 0.19 | **0.04** | -2.02E-07 (-4.94E-07–9.08E-08) | 0.17 | 32 |
| 1 | H106 | 345 | PBMC | 0.29 | 0.58 | 0.14 | -1.38E-07 (-5.08E-07–2.32E-07) | 0.45 | 35 |
| 1 | ZJ15 | 345 | PBMC | 0.71 | 0.99 | 0.81 | -4.85E-09 (-2.71E-07–2.61E-07) | 0.97 | 38 |
| 2 | DFK6 | 260 | PBMC | 0.71 | 0.88 | 0.34 | 3.03E-08 (-1.96E-07–2.56E-07) | 0.79 | 25 |
| 2 | DFR6 | 260 | PBMC | 0.78 | 0.94 | 0.61 | 1.97E-08 (3.53E-07–3.93E-07) | 0.91 | 26 |
| 2 | DG24 | 260 | PBMC | 0.94 | 0.84 | 0.84 | -2.42E-08 (-2.58E-07–2.10E-07) | 0.83 | 32 |
| 2 | GB7V | 260 | PBMC | 0.69 | 0.77 | 0.23 | 4.71E-08 (-1.46E-07–2.40E-07) | 0.62 | 30 |
| 2 | H34G | 260 | PBMC | 0.79 | 0.98 | 0.88 | -1.47E-23 (-2.23E-07–2.23E-07) | 1.00 | 33 |
| 2 | H34H | 260 | PBMC | 0.70 | 0.78 | 0.64 | -5.38E-08 (-3.60E-07–2.52E-07) | 0.72 | 23 |
| 2 | H814 | 260 | PBMC | 0.81 | 0.75 | 0.85 | -2.53E-08 (-2.06E-07–1.55E-07) | 0.78 | 37 |
| 2 | H857 | 260 | PBMC | 0.29 | 0.25 | 0.99 | -1.16E-07 (3.27E-07–9.60E-08) | 0.27 | 31 |
| 2 | H859 | 260 | PBMC | 0.10 | 0.16 | 0.13 | -1.78E-07 (4.12E-07–5.57E-08) | 0.13 | 29 |
| 2 | H860 | 260 | PBMC | 0.72 | 1.00 | 0.39 | 1.47E-08 (-1.57E-07–1.86E-07) | 0.86 | 31 |
| 2 | ZL29 | 260 | PBMC | 0.47 | 0.65 | 0.37 | 5.36E-08 (-1.40E-07–2.47E-07) | 0.58 | 38 |
| 2 | ZM16 | 260 | PBMC | 0.46 | 0.53 | 0.24 | 1.12E-07 (-2.20E-07–4.44E-07) | 0.49 | 24 |
| 2 | All | 260 | PBMC | 0.79 | 0.72 | 0.71 | -1.07E-08 (-7.36E-08–5.22E-08) | 0.74 | 381 |
| 3 | A12V162 | 714 | LN | 0.97 | 0.40 | 0.42 | -1.39E-08 (-5.29E-08–2.51E-08) | 0.48 | 129 |
| 3 | A12V162 | 714 | MES | 0.94 | 0.56 | 0.67 | -9.94E-09 (-5.36E-08–3.37E-08) | 0.65 | 116 |
| 3 | A12V162 | 714 | ALL | 0.99 | 0.32 | 0.39 | -1.21E-08 (-4.47E-08–2.05E-08) | 0.46 | 178 |
| 3 | A12V162 | 1142 | PBMC | 0.10 | 0.31 | 0.07 | 2.09E-08 (-1.68E-08–5.86E-08) | 0.27 | 91 |
| 3 | A12V162E | 714+ | All | 0.62 | 0.69 | 0.54 | 5.31E-09 (-2.19E-08–3.25E-08) | 0.70 | 202 |
| 3 | DW93 | 714 | LN | 0.58 | 0.92 | 0.30 | 3.97E-09 (-5.10E-08–5.89E-08) | 0.89 | 115 |
| 3 | DW93 | 714 | SPL | 0.76 | 0.73 | 0.65 | -1.60E-08 (-1.14E-07–8.15E-08) | 0.74 | 83 |
| 3 | DW93 | 714 | ALL | 0.72 | 0.93 | 0.56 | -1.00E-10 (-5.07E-08–5.05E-08) | 1.00 | 126 |
| 3 | H36J | 714 | LN | 0.41 | 0.13 | 0.40 | -3.21E-08 (-7.42E-08–9.98E-09) | 0.13 | 140 |
| 3 | H36J | 714 | MES | 0.35 | 0.12 | 0.38 | -3.36E-08 (-7.69E-08–9.67E-09) | 0.13 | 136 |
| 3 | H36J | 714 | ALL | 0.29 | 0.05 | 0.11 | -3.28E-08 (-6.59E-08–2.45E-10) | 0.05 | 197 |
| 3 | H36J | 1142 | PBMC | 0.53 | 0.90 | 0.74 | 2.51E-09 (-4.11E-08–4.61E-08) | 0.91 | 111 |
| 3 | H36J | 714+ | All | 0.28 | 0.19 | 0.12 | -1.22E-08 (-4.26E-08–1.83E-08) | 0.43 | 229 |
| 3 | H798 | 714 | LN | 0.92 | 0.98 | 0.86 | 2.41E-10 (-4.26E-08–4.31E-08) | 0.99 | 113 |
| 3 | H798 | 714 | MES | **1.36E-02** | **3.05E-02** | **5.80E-03** | 5.85E-09 (4.45E-09–1.13E-07) | **0.03** | 107 |
| 3 | H798 | 714 | SPL | 0.89 | 0.67 | 0.72 | -8.64E-09 (-5.61E-08–3.88E-08) | 0.72 | 99 |
| 3 | H798 | 714 | ALL | 0.26 | 0.39 | 0.50 | 1.76E-08 (-1.99E-08–5.51E-08) | 0.36 | 183 |
| 3 | H798 | 1142 | PBMC | 0.31 | 0.45 | 0.18 | 1.45E-08 (-2.01E-08–4.91E-08) | 0.41 | 95 |
| 3 | H798 | 714+ | All | 0.22 | 0.31 | 0.50 | 1.59E-08 (-1.48E-08–4.66E-08) | 0.31 | 210 |

A two-sided Wilcoxon rank-sum test was used to assess changes in divergence and randomization tests were used to look for changes in APD and shifts in population structure. The evolutionary rate was estimated using linear regression, with the t-test used to assess if the slope was equal to zero. The p values are not adjusted for multiple comparisons.
[a]P values < 0.05 are highlighted in bold.

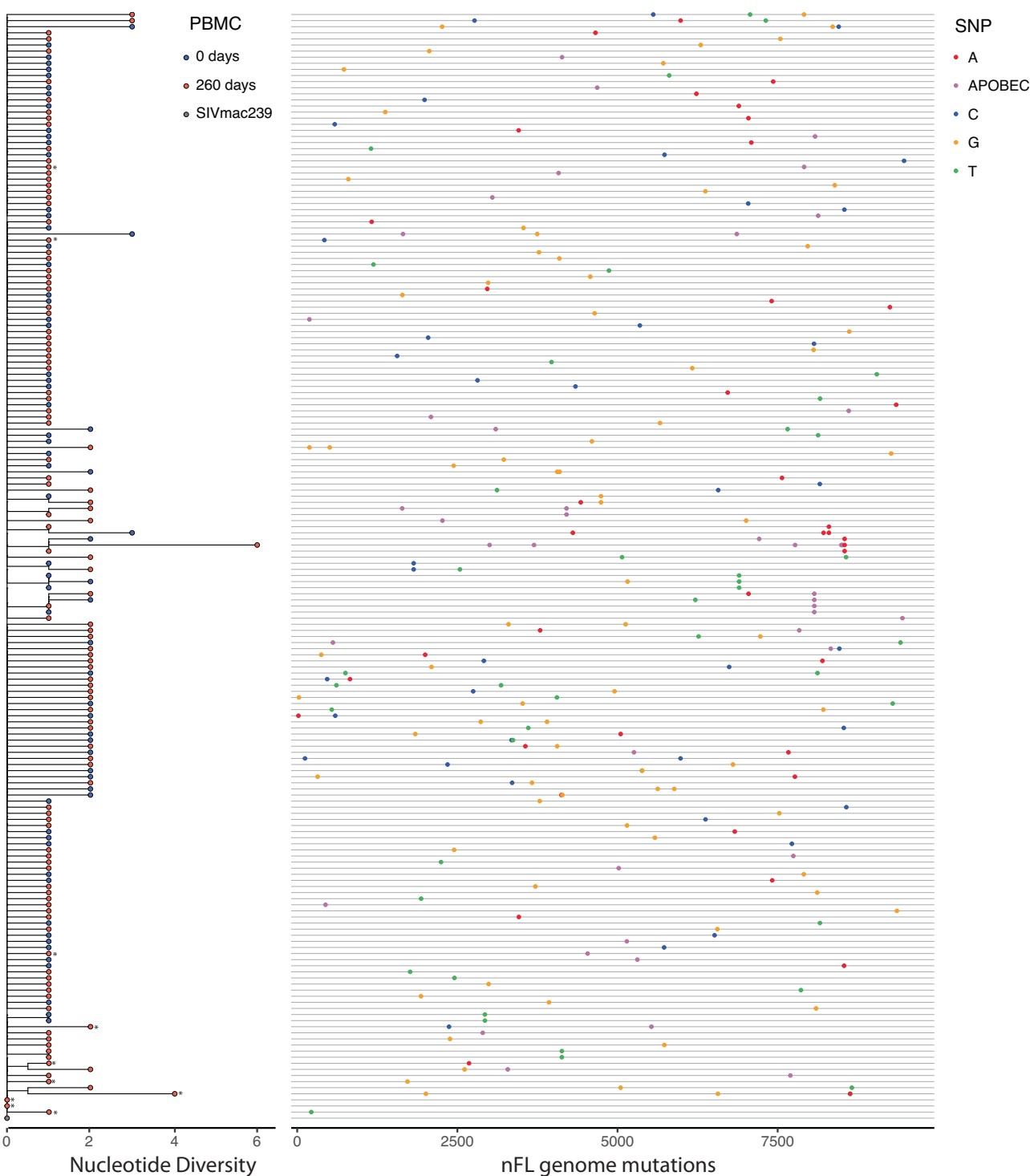

**Fig. 3 | Phylogenetic tree and highlighter alignment show lack of evolution in cohort 2 animals starting ART at 10 dpi.** The neighbor-joining tree combining all nFL SIV DNA sequences obtained from PBMCs for all ten RMs has a star-like structure with few shared mutations between sequences. Single nucleotide polymorphisms (SNP) from the SIVmac239 founder sequence [red for adenine (A), blue for cytosine (C), orange for guanine (G), and green for thymine (T), including potential APOBEC-mediated mutations (purple)] are indicated in the highlighter plot. Asterisks indicate sequences with short indels.

compared to baseline (median APD of 0.013% and *p*-distance of 0.00066% from PBMC; median APD of 0.011% and *p*-distance of 0.00054% from tissues). Genetic diversity declined over time in RM H36J (APD of 0.013% at baseline vs 0.0079% at 714 days on ART, *p* = 0.05; Table 1). Additionally, the tissue and PBMC samples obtained after long-term ART were panmictic with the baseline samples, and the evolutionary rate during ART was not significantly different from zero

in any animal (Fig. 4A). We also repeated the evolutionary analysis after pooling late-ART tissue and PBMC samples to increase statistical power. In line with cohorts 1 and 2, none of the tests for molecular evolution revealed significant changes to the SIV DNA population on ART in any animal (Table 1). The analysis with APOBEC-specific context mutations included did not significantly impact the results (Supplement Table 1).

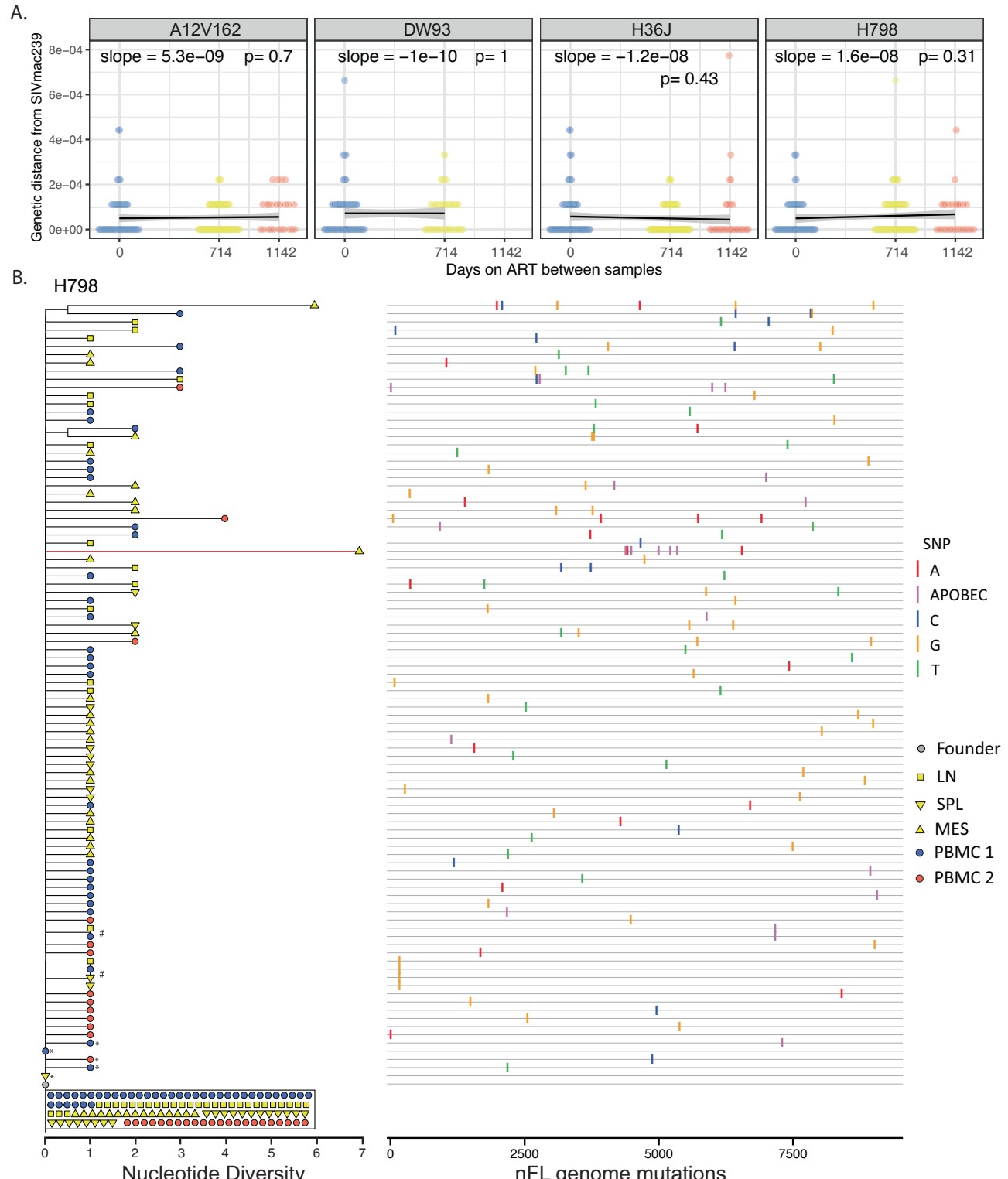

**Fig. 4 | No evidence for evolution in cohort 3 animals initiating ART at 10 dpi.** The scatter plot shows the distribution of p-distances from SIVmac239 at each time point, with black lines corresponding to the linear regression slopes (mutations per site per day) and the grey bands to the 95% confidence intervals around the slopes (**A**). The evolutionary rates during ART are not significantly different from zero in any animal (two-sided *t*-test). The phylogenetic tree for animal H798 shows no evidence of viral evolution during ART (**B**). Clades of potential viral clones (identical sequences with the same genetic barcode) are indicated by hashtag. Red taxa lines indicate APOBEC-induced mutations. Single nucleotide polymorphisms (SNP) from the SIVmac239 founder sequence [red for adenine (A), blue for cytosine (C), orange for guanine (G), and green for thymine (T), including potential APOBEC-mediated mutations (purple)] are indicated in the highlighter plot. Asterisks indicate sequences with short indels. Source data are provided as a Source Data file.

Overall, comparative analysis of longitudinal nFL sequences from tissues and PBMCs did not reveal additional genetic changes during up to 3 years of ART. However, if only a small subset of the virus population had continued to replicate, newly emergent sequences may not have been visible in our population-level analyses. To further probe for the emergence of new variants during ART, we constructed a NJ-phylogeny for each animal and examined if any sequences obtained after long-term ART were on longer branches or were sequestered together on separate clades (Fig. 4B and Supplemental Fig. 4). Overall, the phylogenies were star-like with little branching structure and few shared mutations. Sequences from all time points were interspersed throughout the few clusters in the trees, consistent with a lack of viral evolution during ART. Only 6 out of 775 sequences harbored more than 4 total mutations. Of these six sequences, three included APOBEC signature sites, two had all their mutations clustered within a small region of the viral genome, which is improbable to have occurred during multiple independent replication cycles (simulated $p = 1.7E-6$), and the remaining single sequence contained six authentic, normal nucleotide mutations. This sequence originated in RM H798 from the mesenteric LN (Fig. 4B) that also contained a significantly lower proportion of sequences with zero mutations compared to other tissues and PBMCs in this animal (16/39 vs 113/173, $p = 0.008$, 2-sample test for equality of proportions). This mesenteric LN sample further differed from the earlier PBMC sample in genetic divergence and population structure ($p = 0.041$; Wilcoxon test for divergence; $p = 0.023$; test for panmixia; Holm–Bonferroni (HB)-adjustment of $p$ values for multiple comparisons). However, the extent of genetic changes in mesenteric LN sequences was low overall, with only 0.36 additional mutations per sequence on average compared to those obtained from other tissues and PBMCs (0.82 and 0.46 mutations per sequence, respectively). Overall, our analyses indicate that viral replication before ART initiation can explain the accumulation of mutations on even the longest branches in these phylogenies.

We next assessed if any sequences obtained from the later time point shared an evolutionary history, which could be indicative of continued evolution or selection pressure, even if the overall number of mutations is low. To help distinguish whether two or more sequences sharing a mutation were part of the same lineage or had acquired the mutations independently, we utilized the 34 nt molecular barcode encoded within each viral sequence. Across the 4 phylogenies (Fig. 4B and Supplement Fig. 4), we observed a total of 14 clades sharing a single mutation, 9 of which had the same barcode, indicating that they belonged to the same lineage. In seven of these nine clades, all sequences were identical, suggesting that they may constitute viral sequences in expanded cellular clones. Of these seven sequences, four originated within the same tissue site, one had sequences from two different tissue sites, and two contained sequences from both time points, suggesting that lineages could persist over time and traffic between tissue sites. In addition, the two clades had sequences from both time points that shared the same barcode but were not identical. While the diversification of these lineages is indicative of replication, sequences obtained after long-term ART were not on longer branches than those obtained shortly after ART initiation, suggesting that the observed evolution preceded therapy. The remaining five clades all had sequences with different barcodes, indicating that the mutations had been generated de novo in independent lineages.

We also generated a phylogenetic tree combining sequences from all animals where we found evidence of shared mutations with 14 out of 25 clades containing sequences from two different animals (Supplement Fig. 5). Overall, mutations that emerged in more than one animal did not have distinguishing features from those that were unique to a single sequence. Consistent with previous cohorts, the proportions of nonsynonymous mutations and transversion mutations on non-overlapping coding regions were not significantly different between sequences from the early and late time points (0.67 vs 0.59;

$p = 0.16$, and 0.83 vs 0.87; $p = 0.25$, respectively; two sample $Z$-test of proportions). Overall, our analysis of 775 nFL sequences did not yield any evidence of continued viral evolution in lymphatic tissues or PBMCs during 3 years of suppressive ART.

## No evidence for viral rebound originating from actively replicating lineages at ART discontinuation

Comprehensively characterizing the persisting SIV DNA population in all relevant tissues during ART to rule out a potential sanctuary site permissive to continued replication is not feasible even in NHP models. We, therefore, discontinued ART at 1221 dpi in cohort 3 RMs to allow any actively replicating viral lineages to propagate systemically, utilizing CD8α depletion 7 days after ART discontinuation to increase the number of activated CD4+ T cell targets for expansion[43]. All animals rebounded within 14 days and reached peak viral loads of $6.6 \times 10^4$–$7.0 \times 10^6$ copies/mL by 17 days post-ATI (Fig. 1). We used next-generation barcode sequencing to characterize the composition of the pre-ART and rebounding plasma virus identifying 17 to 28 distinct barcoded viral lineages in each animal at 17 dpi post ART withdrawal. Nearly half of these lineages (48%) were detected in nFL sequences obtained during ART. We plotted all rebounding barcode lineages against the pre-ART rank order of all detectable barcodes (Fig. 5). We observed numerous distinct barcodes (753–1831) pre-ART in each animal. All barcoded lineages detected in rebound viremia were also found in pre-ART plasma, with the majority (60–79%) originating from the top 10% of the pre-ART barcode frequency distribution in each animal. However, several of the rebounding barcode lineages were orders of magnitude lower in frequency at the time of ART initiation than the most abundant lineages, suggesting that they may have expanded while on ART. Overall, the level of replication of individual barcoded lineages at the time of ART initiation was a significant predictor of their subsequent reactivation after ART discontinuation ($p < 1E-8$; logistic regression), but not of their relative contribution to rebound viremia (Pearson Correlation $p = 0.43$–$0.98$).

To assess if any rebounding barcode lineages had accumulated additional mutations during ART, we generated a total of 376 nFL SIV RNA sequences from the rebound plasma of each animal 13 days after ATI. Of these, 11 % ($n = 40$) were excluded from analyses due to having obvious defects (large deletions >300 bp, frameshift mutations, and hypermutations). Additionally, we excluded individual sites where at least one sequence had a potential APOBEC-mediated mutation. We then constructed an NJ-tree for each animal rooted on the SIVmac239 founder sequence and identified the genetic barcode of each viral variant to look for evolution within distinct rebounding lineages (Fig. 6). Using this method, we observed 4–12 different rebounding barcoded lineages per animal all of which were also identified via next-generation sequencing. For three animals, the phylogenies had branching patterns indicative of early evolution within individual lineages, with at least one mutation shared by numerous sequences with the same barcode, whereas animal DW93 had a star-like phylogeny with few shared mutations. Despite several additional days of replication, none of the viral sequences obtained from rebound plasma had more than seven mutations at non-APOBEC sites, comparable to the maximum divergence of sequences obtained during suppressive ART. To assess if the initial virus population that expanded after ART discontinuation had acquired any genetic changes during 3 years of ART, we reconstructed the likely genetic constituents of the initial reactivating variants based on the majority consensus sequence of each barcode lineage detected in rebound plasma. For each rebounding lineage for which we obtained at least two sequences, the presumed founder variant had at most one genetic change compared to SIVmac239, with 60% of initial reactivating variants having no mutations. The predominant lineage in animal H798 provided the clearest example of an initial reactivating variant harboring a single pre-existing mutation, shared by every sequence with the same

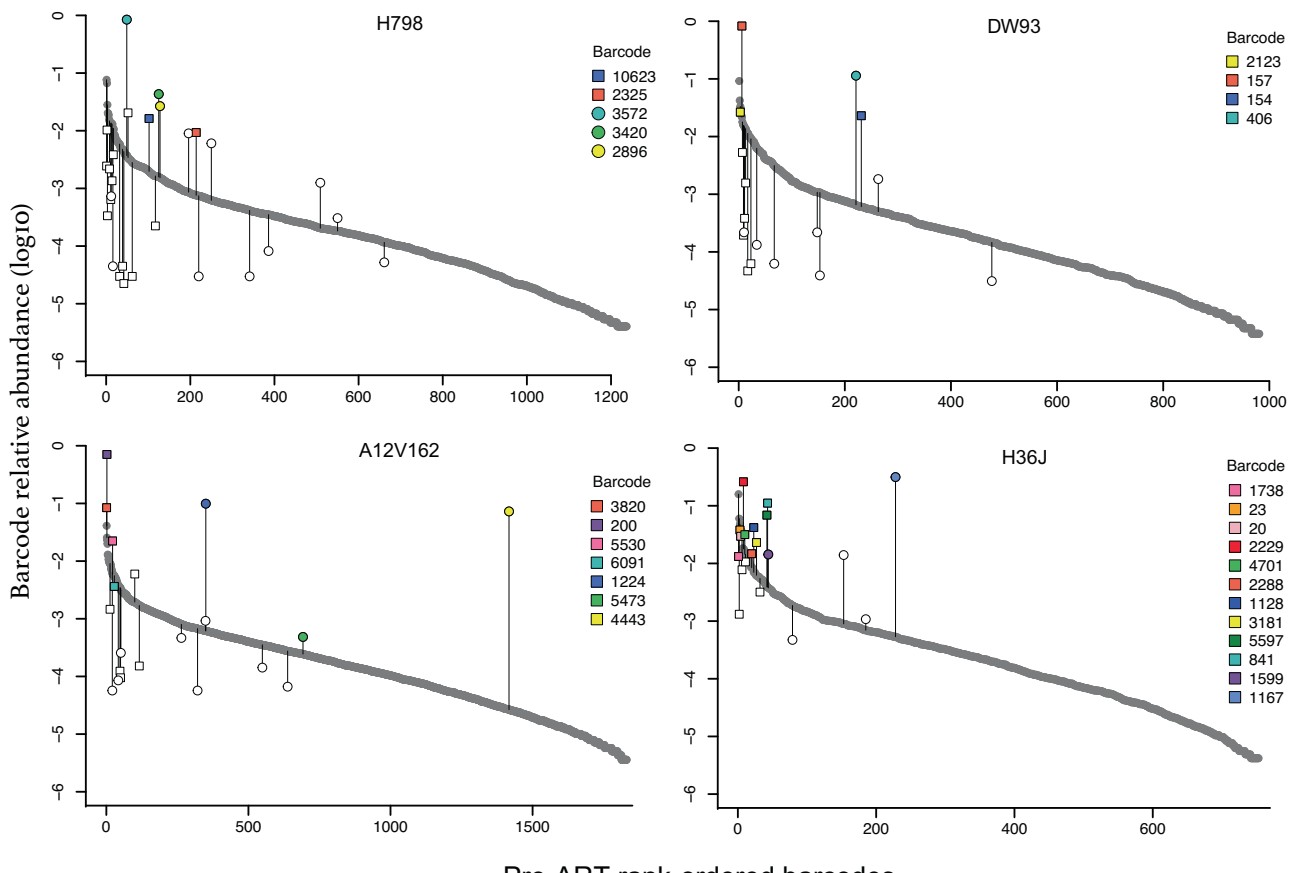

**Fig. 5 | Reactivation of barcoded lineages based on relative abundance in pre-ART plasma.** The grey points depict the relative frequencies (log10) of the rank-ordered barcodes detected in pre-ART plasma. Vertical grey lines highlight which barcodes from the pre-ART distribution were detected following treatment interruption by next-generation barcode sequencing, with their relative frequencies (log10) in rebound plasma viremia indicated by white or colored symbols. Square symbols correspond to barcodes that were detected in at least one on-ART SIV DNA sample, and circles correspond to barcodes that were only detected in pre-ART and off-ART plasma. Individual barcodes that were also found in nFL SIV RNA sequences obtained during ATI are color-coded accordingly. Source data are provided as a Source Data file.

barcode in rebound plasma. None of the barcoded lineages for which we obtained only a single nFL sequence had more than three genetic changes at non-APOBEC sites. Although we could not determine the majority consensus sequences for these lineages due to limited sampling, we expect the initial numbers of genetic changes on the founder variants to be even lower than what was observed on sequences obtained after several days of expansion. Overall, the evolutionary dynamics of the rebounding lineages were consistent with replication after ART discontinuation but not during suppressive ART, indicating that viral rebound originated exclusively from a non-replicating SIV population, thereby refuting the possibility of continued replication in an unsampled tissue sanctuary.

## Discussion

Resolving the question of whether HIV-1 continues to replicate at a low level during suppressive ART, thereby shaping the genetic landscape of the persisting virus population, is critical to the development of treatment strategies that would allow discontinuation of ART without viral recrudescence. Although directly quantifying the level of active HIV-1 replication in tissues is not feasible during ART, the error-prone reverse transcription step of the viral replication cycle introduces genetic changes to the viral genome, imprinting an identifiable signal of evolution on viral sequences. Comparison of viral sequences obtained after long-term ART with those obtained near the time of ART initiation can therefore reveal if the persisting virus population continued to replicate during ART, provided that sampling is deep and comprehensive enough to encompass newly emergent viral variants and additional genetic changes can be distinguished from background genetic diversity. Unfortunately, clinical samples rarely meet both criteria, making it difficult to definitively rule out ongoing cycles of replication during suppressive ART, either at the population level or in an unsampled tissue sanctuary site. To overcome the limitations of clinical sampling, we employed an optimized barcoded NHP model to examine if SIV continued to replicate during long-term ART despite undetectable plasma viremia. Our approach made it possible to (i) limit background genetic diversity, (ii) precisely quantify the divergence of each individual sequence from a known transmitted/founder (T/F) variant, (iii) increase the resolution of analysis to the level of individual mutations within distinct barcoded lineages, (iv) obtain tissue samples during ART, and (v) perform experimental interventions to interrogate the existence of replicating viral populations at unsampled tissue sites. Our analysis of over 1600 nFL sequences from 29 animals did not reveal any additional genetic changes during up to 3 years of ART, apart from single additional mutations in a subset of sequences obtained from the mesenteric LN of one animal.

First, we investigated whether SIV populations in PBMCs had evolved during ART in three cohorts of animals that initiated therapy at 27 dpi or 10 dpi. Following a similar approach as previous studies[18,23,25,30,38], we assessed if SIV DNA obtained after prolonged ART had diverged further from the founder virus, increased in genetic diversity, shifted in population structure, or sequestered phylogenetically compared to samples obtained shortly after ART initiation. In

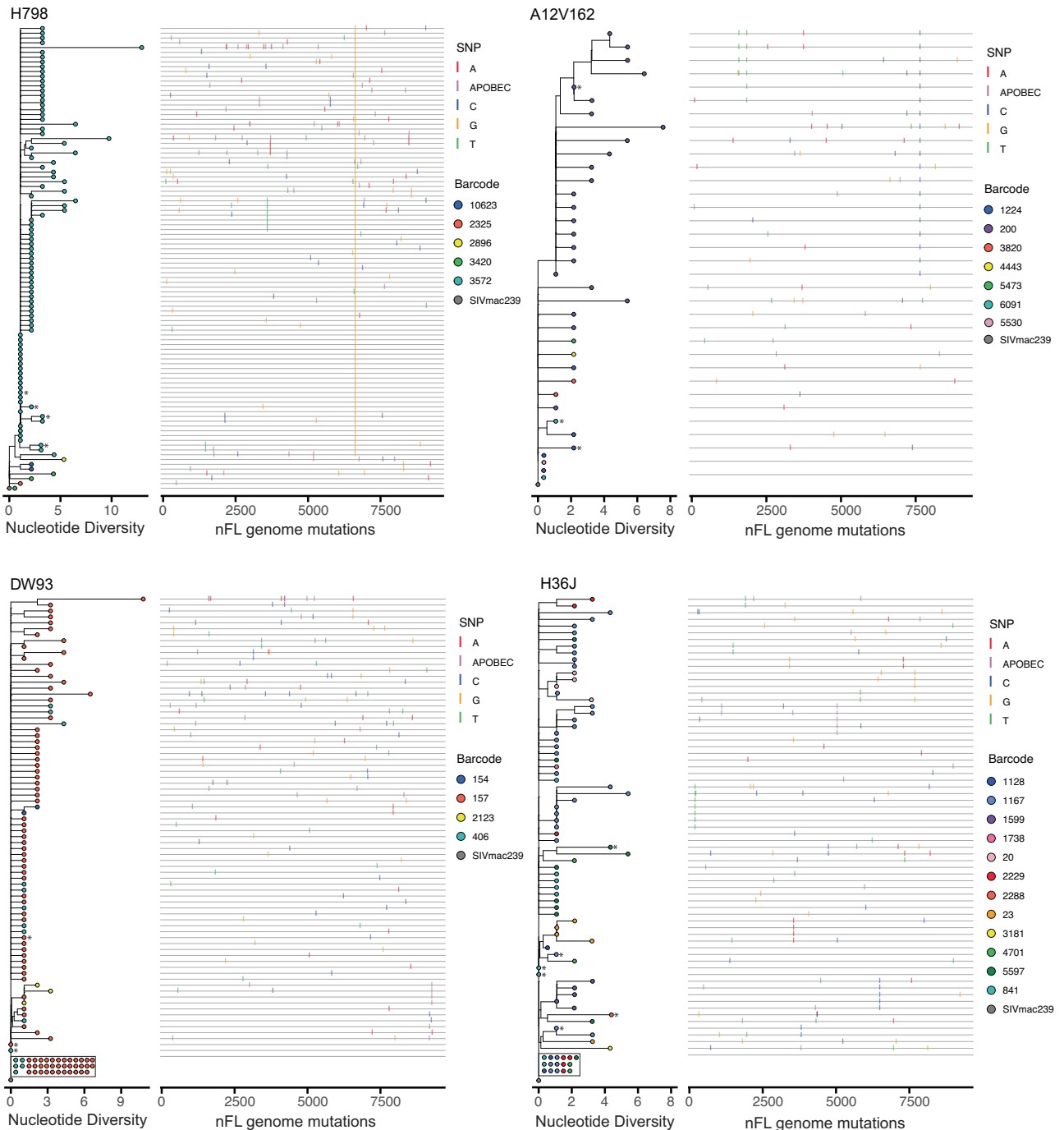

**Fig. 6 | No evidence for rebound from a replicating population for cohort 3 animals.** The color coding of each taxa in the phylogenetic tree indicates the genetic barcode of each nFL SIV RNA sequence obtained from rebound plasma viremia during ATI. Single nucleotide polymorphisms (SNP) from the SIVmac239 founder sequence [red for adenine (A), blue for cytosine (C), orange for guanine (G), and green for thymine (T), including potential APOBEC-mediated mutations (purple)] are indicated in the highlighter plot. Asterisks indicate sequences with short indels.

addition to these population-level analyses, we assessed if any SIV DNA sequences had more genetic changes than expected based on the period of pre-ART replication, and examined individual genetic changes for signs of immune escape or mutational characteristics suggestive of continued selection during ART. None of these analyses yielded any evidence of continued replication during ART in any animal. On the contrary, we observed a trend towards backward evolution in animals that initiated ART at d27, with samples at the later time point having lower genetic divergence and diversity than at baseline in every animal, indicative of the decay of labile virus populations that were

replicating at the time of ART initiation[40]. Overall, these results are consistent with previous studies showing that HIV-1 and SIV populations in PBMCs are stable during ART.

Although virus populations in PBMCs generally reflect those in tissues[38], there have been several reports suggesting ongoing HIV-1 replication restricted to putative sanctuary sites, including lymphatic and gut-associated lymphatic tissues[12,26,28,29,47], despite undetectable virus in the blood. A recent study found that ART concentrations were heterogeneously distributed within lymphatic tissues, with small populations of target cells unexposed to drugs, which could allow for a

few intermittent rounds of replication within these microenvironments, and the emergence of new viral variants with one or two additional mutations prior to the replicating lineages going extinct due to unavailability of targets for expansion[48]. Even if viral replication persists at a continuous, steady-state level in a putative drug sanctuary site leading to the emergence of new virus variants with numerous mutations, detecting them in the persisting SIV DNA population in PBMCs is statistically improbable due to their small proportion of total reservoir size even if they rapidly migrate to the periphery. We, therefore, investigated if SIV DNA populations in lymphatic tissues had continued to evolve during long-term ART in a cohort of 4 animals starting ART at 10 dpi. Sequence analysis of mesenteric LN, axillary LN, and spleen biopsies along with PBMCs after nearly two years of ART as well as PBMC samples from each animal after an additional year of ART did not reveal any evidence of continued viral replication during ART in PBMCs or any sampled tissues.

Importantly, in cohort 3, none of the 775 nFL sequences obtained from tissues and PBMCs during up to 3 years of ART had accumulated more nucleotide changes than could have occurred during 10 days of replication prior to ART initiation, indicating that we did not sample any viral lineages that had been replicating continuously during ART. However, it is not possible to directly rule out the existence of putative tissue sanctuaries during ART without comprehensively assessing all possible sites of replication. As a surrogate for potential lineages replicating actively in unsampled sanctuary sites, we utilized ART discontinuation to allow for systemic spread beyond putative drug sanctuaries with CD8α depletion to facilitate expansion. Despite several additional days of replication during ATI, none of the 336 nFL sequences we obtained from rebound plasma had acquired more mutations than were observed in the pre-ART virus populations, with the rebounding/founder variants of most lineages having zero mutations. The observed lack of evolution in the rebound virus indicates that viral recrudescence originated from the stochastically reactivating viral genomes rather than from an actively replicating virus population. Although we cannot fully exclude the possibility that replication continued during ART in putative sanctuary sites with limited trafficking to the periphery, these viral lineages did not contribute to rebound in this study.

Our observed lack of viral replication during ART suggests that SIV persistence is maintained exclusively by clonal proliferation of long-lived infected cells, which is consistent with HIV-1 clones in people living with HIV[49]. Although we did not specifically investigate clonal expansion in this study, which requires integration site analysis to definitively identify clones, we recovered several sets of identical sequences harboring a single shared nucleotide change and the same genetic barcode. These probable viral clones were sampled from different time points and/or tissue sites, consistent with the observed persistence and systemic distribution of HIV-1[16,23,25]. Additionally, although the level of replication of individual barcoded lineages pre-ART was predictive of whether they subsequently reactivated after ART discontinuation, we observed several rebounding lineages in each animal that were at a low frequency in pre-ART plasma, suggesting that they may have clonally expanded during ART.

Previous studies have shown that virus production from presumed viral clones, rather than ongoing cycles of replication, contributes to a low level of HIV-1 RNA in plasma during ART[50–52], explaining why residual viremia cannot be reduced by treatment intensification[53,54]. While we could not assess sequence evolution in residual SIV viremia due to extremely low viral loads resulting from early ART initiation, our conclusions are in line with the findings from these clinical studies. Although most individuals infected with HIV-1 start treatment much later in chronic infection, the timing of ART initiation is unlikely to influence the ability of the virus to replicate during ART. In fact, the lack of virological failure in individuals adhering to 2-drug treatment regimens provides strong real-world evidence for the ability of ART to completely block ongoing replication during any stage of infection. Overall, our study confirms that the NHP model recapitulates key features of HIV-1 infection, including lack of continued viral replication and the proliferation of infected SIV cells during ART.

Despite the rigorous and comprehensive nature of our analyses, which made it possible to distinguish changes to the SIV DNA population at the level of individual additional mutations in distinct lineages and interrogate the existence of putative drug sanctuaries at unsampled tissue sites, we did not find any evidence for continued replication during ART. Our findings corroborate the growing consensus view that HIV-1 does not replicate during suppressive therapy, support the development of HIV-1 cure strategies that target the non-replicating population of virus that can rebound after ART discontinuation, and highlight the potential utility of the barcoded-NHP model for evaluating the effect of novel curative interventions on the RCVR.

## Methods

### Study design

To look for evidence of ongoing SIV replication during suppressive ART, we used SGS to compare nFL SIV DNA sequences obtained near the time of ART initiation to those obtained after long-term ART in 21 SIV-infected rhesus macaques (RMs) from 3 cohorts that differed in the time of ART initiation and duration of treatment. All RMs were infected intravenously with the barcoded SIVmac239M virus and started ART at 10 (cohort 2 and 3) or 27 (cohort 1) days post-infection (dpi). The genetic diversity of nFL PBMC SIV DNA sequences was compared for between 260–1142 days on ART. Additionally, for cohort 3, we used a laparoscopic procedure to collect mesenteric LN, and spleen in addition to peripheral LN and PBMCs and obtained SIV RNA sequences from rebound plasma after ART discontinuation. Finally, we utilized two untreated Mamu A01+ animals infected with the same virus as positive controls to assess our ability to detect genetic changes occurring during short periods of replication. In these animals, we compared nFL SIV RNA sequences obtained from plasma during active replication.

### Ethics statement

All work involving research animals was conducted under protocols approved by the Animal Care and Use Committee of the National Cancer Institute, and National Institutes of Health (NIH) in NIH-Bethesda facilities. NIH-Bethesda is accredited by AAALAC International and follows the Public Health Service Policy for the Care and Use of Laboratory Animals (Animal Welfare Assurance Number D16-00602). Animal care adhered to the standards outlined in the "Guide for the Care and Use of Laboratory Animals (National Research Council; 2011; National Academies Press; Washington, D.C.), in accordance with the Animal Welfare Act.

### Rhesus macaques

This study used a total of 23 purpose-bred rhesus macaques (*Macaca mulatta*) of Indian genetic background. Animals were evenly divided by sex with 12 females (A12V162, DW93, H798, H34G, H34H, H814, GB7V, H857, ZM16, H859, H860, ZL29) and 11 males (H36J, DEVJ, DEVW, DEVX, H106, ZJ15, DFK6, DFR6, DG24, 7811, 25611) with ages ranging from 3–10 years old at start of study. Due to a lack of detectable evolution in all macaques, comparisons based on sex were not warranted. Animals were specific pathogen-free (cercopithecine herpesvirus 1, D-type simian retrovirus, simian T-lymphotropic virus type 1, rhesus rhadinovirus, and Mycobacterium tuberculosis) prior to study initiation and screened for common Mamu alleles (Mamu-A*01/-A*02 and Mamu-B*08/-B*17) using allele-specific PCR[55]. Only the two control animals had a controlling allele (both Mamu-A*01). Daily ART for

cohort 1 comprised a co-formulated preparation containing the reverse transcriptase inhibitors tenofovir (TFV: (R)-9-(2-phosphonylmethoxypropyl) adenine (PMPA), 20 mg/kg) and emtricitabine (FTC; 50 mg/kg) administered by once-daily subcutaneous injection, plus the integrase strand-transfer inhibitor raltegravir (RAL; 150–200 mg) given orally twice daily (51, 52). Cohort 1 also received ritonavir (RTV)-boosted indinavir (IDV) (120 mg and 100 mg, given orally twice daily) for the first 259 days of treatment, after which IDV and RTV were withdrawn from the regimen. ART for cohorts 2 and 3 consisted of a single subcutaneous injection of co-formulated 5.1 mg/kg tenofovir disoproxil (TDF), 40 mg/kg emtricitabine (FTC) and 2.5 mg/kg dolutegravir (DTG) combined in a solution containing 15% (w/w) Kleptose at pH 4.2[44,56].

### Viruses

The barcoded virus model (SIVmac239M) was utilized in this study. SIVmac239M is a virus stock generated from the original Nef-open SIVmac239 clone with a short genetic insertion between the *upx* and *upr* genes[41,45]. This 34-base genetic insert contains 10 random bases providing over 1 million possible unique sequences representing a viral barcode. The barcode can be sequenced by next-generation sequencing thereby generating a genetically diverse virus population that is isogenic outside the viral barcode. In this study, we used a high dose of SIVmac239M ranging from 10,000 IU (cohort 3) to 220,000 IU (cohorts 1 and 2) as determined by TZMbl assay.

### Barcode sequencing

RNA isolation from plasma was performed using a QIAamp Viral RNA mini kit (Qiagen). Complementary DNA (cDNA) was generated with Superscript III reverse transcriptase (ThermoFisher) and an SIV-specific reverse primer (Vpr.cDNA3: 5′-CAG GTT GGC CGA TTC TGG AGT GGA TGC-3′). The cDNA was quantified via qRT-PCR using the primers VpxF1 5′-CTA GGG GAA GGA CAT GGG GCA GG-3′ and VprR1 5′-CCA GAA CCT CCA CTA CCC ATT CATC with labeled probe (ACC TCC AGA AAA TGA AGG ACC ACA AAG GG). Prior to sequencing, PCR was performed with VpxF1 and VprR1 primers containing either the F5 or F7 Illumina adapters each with a unique 8-nucleotide index sequence for multiplexing. PCR was performed using High Fidelity Platinum Taq (ThermoFisher). The multiplexed samples were sequenced on a MiSeq instrument (Illumina) and analyzed as described below.

### Viral detection assays

RNA isolation from plasma was performed using QIAgen DSP virus/pathogen Midi kits (Qiagen) on the QIASymphony-XP instrument (Qiagen). Complementary DNA (cDNA) was generated with Superscript III reverse transcriptase (ThermoFisher) and random hexamers. Real-time quantitative PCR was performed using forward primer (SGAG21), 5′-GTC TGC GTC ATP TGG TGC ATTC-3′; reverse primer (SGAG22), 5′-CAC TAG KTG TCT CTG CAC TAT PTG TTT TG-3′ and probe (pSGAG23), 5′-(FAM)-CTT CPT CAG TKT GTT TCA CTT TCT CTT CTG CG-(BHQ)-3′, with TaqGold polymerase (ThermoFisher). This assay has a 15 copy/mL limit of detection[57,58].

### Near full-length single genome sequencing

Viral genome sequencing was performed by single genome amplification followed by direct Sanger sequencing from CA-DNA obtained from PBMCs and described tissues. Limiting dilution PCR was performed using Platinum SuperFi DNA polymerase (Invitrogen) and the following primers: SIVnFL.F1 (5′-GAT TGG CGC CYG AAC AGG GAC TTG-3′) and SIVnFL.R1 (5′-CCC AAA GCA GAA AGG GTC CTA ACG-3′), followed by a second round PCR using the following primers: SIVnFL.F2 (5′-GTG AAG GCA GTA AGG GCG GCA GG-3′) and SIVnFL.R2 (5′-CCA GGC GGC GRC TAG GAG AGA TGG-3′). All nFL sequence data in this study have been deposited in the GenBank database under accession code PP486481-PP488222.

### Near full-length sequence alignment and quality control

Sequences from each animal were aligned using ClustalW and the alignments were manually edited around gap regions in AliView. Codon-aligned nucleotide sequence alignments were constructed for each gene using the Gene Cutter tool from the Los Alamos HIV Sequence Database. The exclusion criteria included sequences with large deletions >300 bp, frameshift and stop codon mutations, and hypermutated sequences ($p < 0.05$). Hypermut tool was used to detect APOBEC-induced hypermutation[59]; https://www.hiv.lanl.gov.

### Genetic and statistical analysis

Neighbor-joining trees were constructed individually for each animal in cohorts 1, 2, and 4 and jointly for cohort 3 animals. For analyses of molecular evolution, we masked all nucleotide positions where at least one sequence in a given alignment had a mutation at an APOBEC signature site, identified by the Highlighter tool found at LANL HIV website (https://www.hiv.lanl.gov). The analyses were also repeated for all nucleotide positions, presented in the supplement. The genetic divergence of each sequence from the SIVmac239 founder lineage was defined as the proportion of sites with nucleotide substitutions, ignoring deletions and indels (*p*-distance). The 2-tailed Mann–Whitney *U* test was used to assess if the p-distances of the viral sequence populations were significantly different between the early and late time points in each animal. Linear regression was used to estimate the slope of *p*-distance as a function of time in each animal, with the *t*-test used to determine if the slope was significantly different from zero. The APD between sequences in each sample was computed as $\frac{n}{n-1}\sum_{i=2}^{n}\sum_{j=1}^{i-1}2x_ix_j\pi_{ij}$, where $x_i$ and $x_j$ are the respective frequencies of the *i*th and *j*th sequences, $\pi_{ij}$ is the number of nucleotide differences between them and $n$ is the number of sequences in the sample. A randomization test with 10,000 iterations was used to assess if the change in APD between early and late samples was significantly different from zero (null hypothesis) for each animal, with the *p*-value estimated as the proportion of the randomized samples that were larger than the observed test static (absolute difference in APD) was greater than or equal to the observed value. A randomization test for panmixia was used to look for a shift in population structure between samples, as described in ref. 25. Specifically, the *p*-value was estimated as the proportion of the 10,000 randomized samples that were smaller than or equal to the observed test statistic $K_s^*$, defined as $K_s^* = w_1 K_1^* + w_2 K_2^*$, with weights $w_1 = \frac{n_1-2}{n_1+n_2-4}$ and $w_2 = 1 - w_1$. Here $n_1$ and $n_2$ are the number of sequences in first and second sample, respectively, and $K_i^* = \sum_{a=1}^{n_i-1}\sum_{b=a+1}^{n_i}\log(1+D_{ab})/\binom{n_i}{2}$, where $D_{ab}$ is the number of differences between sequence $a$ and $b$ of sample $i$. All statistical and genetic analyses were performed in R v4.3.1. BioStrings v3.19, Ape v5.8, and GGtree v3.8.2 packages were used for processing and visualizing sequence data.

### Reporting summary

Further information on research design is available in the Nature Portfolio Reporting Summary linked to this article.

## Data availability

All nFL sequence data in this study have been deposited in the GenBank database under accession code PP486481-PP488222 (https://www.ncbi.nlm.nih.gov/popset/?term=2701657060). The processed data generated in this study are provided in the Supplementary Information/Source Data file. Source data are provided with this paper.

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

## Acknowledgements

We are grateful and acknowledge ViiV Healthcare for DTG, Gilead Sciences Inc. for TFV and FTC, and Merck for RAL and IDV. We thank the Quantitative Molecular Diagnostics Core of the AIDS and Cancer Virus Program of the Frederick National Laboratory for viral load analyses, George Muthua and the animal care staff of the Laboratory Animal Sciences Program, Frederick National Laboratory, for expert animal care and support, and the Nonhuman Primate Research Support Core of the AIDS and Cancer Virus Program for critical specimen processing, logistic support, and ART coformulation. This project has been funded in whole or in part with federal funds from the National Cancer Institute, National Institutes of Health, under Contract No. 75N91019D00024/HHSN261201500003I. The content of this publication does not necessarily reflect the views or policies of the Department of Health and Human Services, nor does mention of trade names, commercial products, or organizations imply endorsement by the U.S. Government.

## Author contributions

Conceptualization: B.F.K., T.T.I. Methodology: B.F.K., C.M.F., G.Q.D.P., J.D.L., T.T.I. Investigation: A.M., B.F.K., C.M.F., J.D.L., L.L., L.N., M.B., N.W. Formal Analysis: B.F.K., T.T.I. Writing—original draft: B.F.K., T.T.I. Writing—review and editing: B.F.K., G.Q.D.P., J.D.L., T.T.I.

## Competing interests

The authors declare no competing interests.

## Ethical approval

All applicable institutional and national guidelines for the care and use of animals were followed.
