## [Peer Review File · Nature Communications]

No evidence for ongoing replication on ART in SIV infected macaquesReviewers' Comments:

Reviewer #1:

Remarks to the Author:

Summary: It is well known that ART treatment for HIV infection is effective, yet individuals who come off such treatment often have an increase in viral load potentially leading to AIDS. A critical question in such rebound cases is whether the viral population is actively replicating at low levels, effectively restocking viral reservoirs or if the viral populations are dormant in these reservoirs waiting for the end of ART to allow subsequent replication. This paper effectively answers this question in the macaque model by taking advantage of this model to inoculate with a known viral strain and track evolution (or not) during and after ART. The study is well conceived and executed and the results highly relevant. There are limitations to the model, of course, but the study provides strong evidence that reservoirs are not 'restocked' by low-level viral replication. The study data are publicly available. I would recommend the authors make the analytical scripts available as well. I've noted a few issues below where I think the authors could make things a bit more clear and tidy up the writing (there is a lot of duplication and the paper is pretty long). In general, the results are well supported by the data and associated analyses.

Abstract – define NHP at first usage.

The authors suggest three approaches for detecting molecular evolution after ART, namely, genetic divergence from the 'T/F sequence' (although 'T/F' is never actually defined, genetic diversity, population genetic structure, and evolutionary rate different from zero. I would argue these are all good measures for evolution except population structure (test for panmixia). This has nothing to do with evolution over time, but with substructuring of evolution. This does not seem particularly relevant to the question at hand – are the viral genomes evolving. It would be a next step if the answer was 'yes', but the answer seems to be 'no'. So I would recommend the authors not pitch this as a 'test' for detecting molecular evolution in their data. Actually, now that I'm in the results section, the authors are using the panmixia test by treating the starting time point and subsequent time point(s) as separate 'populations' and asking if they are different. This seems fine to me, but you need to better explain this in the methods section. Also, there are a lot of 'methods' being outlined in the introduction. I would argue the authors should streamline the introduction and keep the methods in the methods section and results in the results section.

Were alignments really manually edited? It seems like with variation this low, you would not need to do such things. Also, Clustalw has been shown repeatedly to be far inferior (both in speed and accuracy) to iterative approaches such as MAFFT. If you used a better alignment tool, you would not have to do 'manual editing'. How many 'hypermutated' sequences were actually removed from the data set? What's the justification for removing these lineages that seem to give the appearance of having undergone evolution? (OK, never mind, I see this in the results now [lines 115-117]. You might mention something in the methods about this as well – exclusion criteria).

Analytical R scripts should be made available (open access) via GitHub or some such accessible repository.

Thank you for making the sequence data publicly available. I have verified that the sequences are accessible on GenBank. I see they are labeled 'cohort 1', 2 and 3 respectively which is great and easily relates back to Table 1. However, are the control sequences also included in GenBank and Table 1? It seems not. The authors should include these data as well. Also, it would be ideal if the authors either put together a PopSet or BioProject to link these sequences together.

Reviewer #2:

Remarks to the Author:

Immonen and colleagues conducted a thorough analysis of SIV replication after ART by combining several monkey cohorts, all infected with barcoded SIVmac239M, using various ART regimens initiated early after infection. Collectively, Immonen et al intravenously infected 21 rhesus macaques (RMs) with barcoded SIVmac239M, and initiated ART between 10 and 27 days after infection. Suppressive ART was maintained for 285-1200 days. Near full-length sequences were generated from blood or tissues (LN and spleen in cohort 3). This sequencing data set was then interrogated using standard evolutionary tests applied to a total of 1,363 SIV DNA near full length single genome sequences. Analysis of rebound virus after ART stop was also examined in cohort 3. Collectively, the results demonstrate that sequences obtained after long-term ART had not diverged significantly from founder sequences initiating infection in blood or the tissues studied. This work adds to a significant literature, cited by the authors, indicating that no apparent viral replication is ongoing during suppressive antiretroviral therapy.

Overall, these studies were well-conducted with technical precision by a group that has significant experience in this area. I have no major criticisms of the study design or technical nature as all of the studies and analysis conducted here are routine in the field, and certainly routine in the laboratory of the senior author. My only major concern is that these analyses, while quite intensive, offer limited new insight to this area. The authors also acknowledge this same point throughout the manuscript and largely confirm existing dogma within the field.

Reviewer #3:

Remarks to the Author:

The authors of this excellent manuscript have addressed in a rigorous way an important question in the HIV field that has largely already been settled. The question is whether antiretroviral therapy (ART) completely blocks viral replication. In an SIV model in which animals are treated shortly after infection, they find no evidence for ongoing cycles of replication. I have only a few suggestions for improving the manuscript.

- 1) The authors should point out more clearly that the animals were treated shortly after infection, while most infected individuals start treatment during chronic infection. It is not clear that the results would be any different, but the authors could discuss this possibility.
- 2) The authors cite a controversial paper (ref 27) that claimed viral evolution during ART. It should probably be mentioned that that paper was challenged on methodological grounds.
- 3) The authors did not look for evolution in the residual viremia, the trace level of free virus in the blood of treated patients. If evolution was occurring this would be a good place to look since this virus is actually being produced and released. The authors should cite papers showing a lack of evolution in the residual viremia and papers showing that it cannot be reduced by treatment intensification. This work provides strong support for the conclusions in this paper.
- 4) The authors should discuss the current HIV treatment landscape, especially the recent changes in treatment guidelines that allow switching from 3 drug regimens to simpler 2 drug regimens. This is powerful real-world evidence with a very large n for the ability of ART to completely block replication and evolution in adherent patients.

Reviewer #1 (Remarks to the Author):

Summary: It is well known that ART treatment for HIV infection is effective, yet individuals who come off such treatment often have an increase in viral load potentially leading to AIDS. A critical question in such rebound cases is whether the viral population is actively replicating at low levels, effectively restocking viral reservoirs or if the viral populations are dormant in these reservoirs waiting for the end of ART to allow subsequent replication. This paper effectively answers this question in the macaque model by taking advantage of this model to inoculate with a known viral strain and track evolution (or not) during and after ART. The study is well conceived and executed and the results highly relevant. There are limitations to the model, of course, but the study provides strong evidence that reservoirs are not 'restocked' by low-level viral replication. The study data are publicly available. I would recommend the authors make the analytical scripts available as well. I've noted a few issues below where I think the authors could make things a bit more clear and tidy up the writing (there is a lot of duplication and the paper is pretty long). In general, the results are well supported by the data and associated analyses.

Abstract – define NHP at first usage.

The authors suggest three approaches for detecting molecular evolution after ART, namely, genetic divergence from the 'T/F sequence' (although 'T/F' is never actually defined, genetic diversity, population genetic structure, and evolutionary rate different from zero. I would argue these are all good measures for evolution except population structure (test for panmixia). This has nothing to do with evolution over time, but with substructuring of evolution. This does not seem particularly relevant to the question at hand – are the viral genomes evolving. It would be a next step if the answer was 'yes', but the answer seems to be 'no'. So I would recommend the authors not pitch this as a 'test' for detecting molecular evolution in their data. Actually, now that I'm in the results section, the authors are using the panmixia test by treating the starting time point and subsequent time point(s) as separate 'populations' and asking if they are different. This seems fine to me, but you need to better explain this in the methods section. Also, there are a lot of 'methods' being outlined in the introduction. I would argue the authors should streamline the introduction and keep the methods in the methods section and results in the results section.

Were alignments really manually edited? It seems like with variation this low, you would not need to do such things. Also, Clustalw has been shown repeatedly to be far inferior (both in speed and accuracy) to iterative approaches such as MAFFT. If you used a better alignment tool, you would not have to do 'manual editing'

What's the justification for removing these lineages that seem to give the appearance of having undergone evolution? (OK, never mind, I see this in the results now [lines 115-117]).

Analytical R scripts should be made available (open access) via GitHub or some such

accessible repository.

Thank you for making the sequence data publicly available. I have verified that the sequences are accessible on GenBank. I see they are labeled 'cohort 1', 2 and 3 respectively which is great and easily relates back to Table 1. However, are the control sequences also included in GenBank and Table 1? It seems not. The authors should include these data as well. Also, it would be ideal of the authors either put together a PopSet or BioProject to link these sequences together.

We appreciate the reviewer's careful reading of our paper and thoughtful suggestions. We have streamlined the introduction to remove methods as suggested and included the missing definitions of acronyms. We have also expanded the Methods section to include a statement on the exclusion criteria (with more details on detection of significantly hypermutated sequences, which do not follow a standard evolutionary process), providing a detailed description of the test for a shift in population structure and our statistical methods overall. We thank the reviewer for the suggestion of using Mafft in future work and have further clarified in the Methods that the alignments were manually edited around gap regions (which was a minimal effort). We also thank the reviewer for noticing our oversight in not including the results for the control animals in Table 1 and Supplement Table 1, which we have now been added. They are included in the GenBank PopSet with all other cohort sequences. We have limited our analyses to standard methods in R.

Reviewer #2 (Remarks to the Author):

Immonen and colleagues conducted a thorough analysis of SIV replication after ART by combining several monkey cohorts, all infected with barcoded SIVmac239M, using various ART regimens initiated early after infection. Collectively, Immonen et al intravenously infected 21 rhesus macaques (RMs) with barcoded SIVmac239M, and initiated ART between 10 and 27 days after infection. Suppressive ART was maintained for 285-1200 days. Near full-length sequences were generated from blood or tissues (LN and spleen in cohort 3). This sequencing data set was then interrogated using standard evolutionary tests applied to a total of 1,363 SIV DNA near full length single genome sequences. Analysis of rebound virus after ART stop was also examined in cohort 3. Collectively, the results demonstrate that sequences obtained after long-term ART had not diverged significantly from founder sequences initiating infection in blood or the tissues studied. This work adds to a significant literature, cited by the authors, indicating that no apparent viral replication is ongoing during suppressive antiretroviral therapy.

Overall, these studies were well-conducted with technical precision by a group that has significant experience in this area. I have no major criticisms of the study design or technical nature as all of the studies and analysis conducted here are routine in the field, and certainly routine in the laboratory of the senior author. My only major concern is that these analyses, while quite intensive, offer limited new insight to this area. The

authors also acknowledge this same point throughout the manuscript and largely confirm existing dogma within the field.

We appreciate the reviewer's careful reading of our manuscript and are happy with their assessment of our work as technically rigorous, intensive, and well-conducted. We agree that our conclusions, which support the consensus view of a lack of continued replication on ART, do not offer new fundamental insights to the field. However, we feel that the greater sensitivity and comprehensiveness of our analyses allowed us to address the question more definitively than has been possible with clinical samples and importantly, confirms the suitability of nonhuman primate models for cure research.

Reviewer #3 (Remarks to the Author):

The authors of this excellent manuscript have addressed in a rigorous way an important question in the HIV field that has largely already been settled. The question is whether antiretroviral therapy (ART) completely blocks viral replication. In an SIV model in which animals are treated shortly after infection, they find no evidence for ongoing cycles of replication. I have only a few suggestions for improving the manuscript.

1) The authors should point out more clearly that the animals were treated shortly after infection, while most infected individuals start treatment during chronic infection. It is not clear that the results would be any different, but the authors could discuss this possibility.

2) The authors cite a controversial paper (ref 27) that claimed viral evolution during ART. It should probably be mentioned that that paper was challenged on methodological grounds.

3) The authors did not look for evolution in the residual viremia, the trace level of free virus in the blood of treated patients. If evolution was occurring this would be a good place to look since this virus is actually being produced and released. The authors should cite papers showing a lack of evolution in the residual viremia and papers showing that it cannot be reduced by treatment intensification. This work provides strong support for the conclusions in this paper.

4) The authors should discuss the current HIV treatment landscape, especially the recent changes in treatment guidelines that allow switching from 3 drug regimens to simpler 2 drug regimens. This is powerful real-world evidence with a very large n for the ability of ART to completely block replication and evolution in adherent patients.

We greatly appreciate the reviewer's insightful comments and suggestions that have helped us better contextualize our findings considering previous work and clinical relevance. We described the controversy around the methodological shortcomings of the Redondo et al. paper in the Introduction and included a paragraph discussing the implications of early ART initiation in our animals vs individuals with HIV-1 infection, lack of evolution in residual viremia, and the success of 2-drug regimens in the Discussion.